# Wavelet-Based Output-Only Damage Detection of Composite Structures

**DOI:** 10.3390/s23136121

**Published:** 2023-07-03

**Authors:** Rims Janeliukstis, Deniss Mironovs

**Affiliations:** Institute of Materials and Structures, Riga Technical University, LV-1048 Riga, Latvia; deniss.mironovs@rtu.lv

**Keywords:** statistical damage detection, wavelet transform, modal features, composite structure

## Abstract

Health monitoring of structures operating in ambient environments is performed through operational modal analysis, where the identified modal parameters, such as resonant frequencies, damping ratios and operation deflection shapes, characterize the state of structural integrity. The current study shows that, first, time-frequency methods, such as continuous wavelet transform, can be used to identify these parameters and may even provide a large amount of such data, increasing the reliability of structural health monitoring systems. Second, the identified resonant frequencies and damping ratios are used as features in a damage-detection scheme, utilizing the kernel density estimate (KDE) of an underlying probability distribution of features. The Euclidean distance between the centroids of the KDEs, at reference and in various other cases of structural integrity, is used as an indicator of deviation from reference. Validation of the algorithm was carried out in a vast experimental campaign on glass fibre-reinforced polymer samples with a cylindrical shell structure subjected to varying degrees of damage. The proposed damage indicator, when compared with the well-known Mahalanobis distance metric, yielded comparable damage detection accuracy, while at the same time being not only simpler to calculate but also able to capture the severity of damage.

## 1. Introduction

Structures made of conventional materials are being extensively replaced by fibre-reinforced polymer (FRP) composite materials in the aerospace, automotive, energy and other industries owing to their superior specific strength [1], negligible thermal expansion, as well as fatigue and corrosion resistance compared to metals. The safety and reliability of structures, such as aircraft fuselages, helicopter blades, as well as wind turbine blades, is ensured by surveys using non-destructive testing (NDT) methods or planned maintenance. Many NDT approaches require manual inspection, whereas planned maintenance often assumes taking the structure out of operation, which increases downtime and increases costs. What is more is that sometimes the inspection is unnecessary due to lack of damage. In order to detect the onset and progression of existing structural damage in a timely manner and reduce the maintenance costs, effective structural health monitoring (SHM) solutions are crucial.

Output-only SHM involves the detection and possible characterization of damage by analysing only the response signals collected from sensors mounted on the structure. Structural excitation can be provided if conditions for operational modal analysis (OMA) are fulfilled [2]. In practice, such is the case of ambient excitation of a stochastic nature, for example, that is caused by wind, sea waves or traffic loads. Specialized signal-processing algorithms can be applied to these output-only responses to extract essential information on the status of structural integrity. This is normally achieved through a statistical pattern recognition framework, where damage-sensitive features (DSFs) are extracted from the sensor measurements [3]. The DSFs in OMA are typically modal parameters—resonant frequencies, damping ratios and mode shapes. Although several techniques of modal parameter identification exist, such as stochastic subspace identification [4] and least-squares complex exponential [5], etc., it is possible to extract only a limited amount of modal parameter observations and only from repeated measurements. For a statistical damage-detection approach using a significantly larger amount of data, traditional OMA modal parameter estimation techniques can be substituted by time-frequency analysis.

Time-frequency analysis methods, such as short-time Fourier transform, Wigner–Ville distribution and Hilbert–Huang transform and wavelet transform (WT) [6], among others, have become established techniques for analysing transient signals, which are nonstationary in nature. In the case of OMA, free structural vibrations induced by ambient excitations are transient signals with a finite energy localized in time and frequency. Another merit of time-frequency techniques is that they have an ability to decompose a composite signal consisting of several modes of vibration (degrees of freedom) into individual modes [7]. This is normally achieved by finding wavelet ridges—high energy curves in a time-frequency plane tracing that allow for system identification via the extraction of modal parameters. Staszewski was the first to demonstrate the WT can be used as a tool for structural modal parameter identification [8]. Wavelet transform has been used in modal parameter estimation for real structures, such as long-span cable-stay bridges and suspension bridges using continuous wavelet transform in [9]; a cable-stay bridge in Taiwan using wavelet packet transform in [10]; a 600 m tall building in China using a combination of empirical wavelet transform and Hilbert transform in [11]; and a pedestrian overpass in the USA using multisynchrosqueezing transform, a variation of wavelet transform that yields a more concentrated estimate of modal parameters at the cost of higher computational complexity in [12]. Authors in [13] proposed an output-only modal identification and structural damage detection technique based on time-frequency techniques, including wavelets. The above studies have focused on modal parameter estimation via wavelet transform. However, it has been demonstrated in [14] that the time-frequency approach with continuous wavelet transform (CWT) allows for the extraction of numerous instances of modal parameters which, when employed in statistical pattern recognition schemes, are more beneficial, since more data are available. This leads to a larger dataset and, therefore, issues of model overfitting and underfitting can be solved.

After the extraction of DSFs, an anomaly detection algorithm is employed to identify outliers supposedly originating from damage or changes in environmental conditions [15]. A popular class of methods of anomaly detection is based on dissimilarities between a reference structural state and a potentially anomalous state. The Mahalanobis distance (MD) metric has been successfully used for such purposes [16,17,18], owing to its ability to detect outliers in a multidimensional feature space with an arbitrary number of extracted DSFs. However, for MD to be used, DSFs have to follow a normal distribution. On the other hand, kernel density estimation (KDE) is an approach used to identify the underlying probability density function without any assumptions regarding a probability distribution of data. KDE was used for structural damage detection in [19,20].

Building on the concept introduced in [14], the aim of the present study was to develop a structural damage detection algorithm using continuous wavelet transform as an alternative modal parameter estimation technique. The use of CWT enables a statistical approach to damage detection using KDE. The underlying tasks of the study were to estimate the probability density function of the extracted modal parameters and calculate the probability centroids for modal features. Finally, Euclidean distances between centroids at a reference point and various states of damage were to be calculated, explored as a potential damage indicator and compared with a Mahalanobis distance in terms of accuracy.

## 2. Modal Identification

The current study is based on the concepts described in detail in [14]. Wavelet ridges are hidden constituent elements of a finite-energy signal revealed through wavelet decomposition of a said signal. Ridges contain all of the essential information on structural modal parameters of a structure whose impulse response is available. Wavelet phase can be used to extract numerous observations of resonant frequencies and damping ratios for each mode of structural vibration. These instances comprise a dataset that is representative of a structural condition in the current state. Each new structural state, for example, the occurrence of damage is associated with changes in modal parameter values. This concept can be utilized in machine-learning-aided structural damage detection and, in a broader sense, structural health monitoring (SHM).

A quick recap of the methodology from [14] is given as follows:


CWT on the recorded response signals $y\left(t\right)$ is carried out using analytical Morlet function and storing complex-valued CWT coefficients in a matrix form:(1)Wy=ReWys1, b1−iImWys1, b1…ReWys1,bL−iImWys1,bL⋮⋱⋮ReWysn,b1−iImWysn,b1…ReWysn,bL−iImWysn,bL,
where $s={\left[\begin{array}{ccc} s_{1} & \ldots & s_{n} \end{array}\right]}^{T}$ is the vector of scale factors, $b={\left[\begin{array}{ccc} b_{1} & \ldots & b_{L} \end{array}\right]}^{T}$ is the vector of translation parameters, and L is the signal length.Wavelet ridges (in terms of scale parameters $s_{i}^{*}$) of each response signal are found by, firstly, finding the s and b parameters (denoted by $s^{*}$ and $b^{*}$) corresponding to the maximum value of modulus of CWT coefficients and, secondly, testing the ridge condition at a fixed parameter $b^{*}$ (time instant when vibration amplitude is maximum).




(2)
$$\frac{d}{ds}W_{y}\left(s,\;b^{*}\right)=0.$$



The conversion from scale parameter to frequency is performed through the following relation:(3)$$f=\frac{1}{2\pi }\times \frac{\omega_{0}}{s},$$
where $\omega_{0}$ is the central frequency of wavelet function. It is essentially a pseudofrequency or a frequency that the wavelet function would have if it was a harmonic function.

Damped natural frequencies are calculated from the derivative of phase between the real and imaginary parts of CWT coefficients along the wavelet ridge line with respect to time. The wavelet ridge line is defined at the ridge scales $s_{i}^{*}$ along the whole time span of free vibrations starting from the time instant $b^{*}$.



(4)
$$\frac{d}{dt}\mathrm{Arg}\left(W_{y}\left(s_{i}^{*},b^{*}:b_{L}\right)\right)=\omega_{d}\frac{d}{ds}W_{y}\left(s,\;b^{*}\right)=0.$$



Damping ratios are extracted in the following substeps:

The moduli of CWT coefficients are extracted along the wavelet ridge, and its natural logarithm is calculated. It is denoted as $\ln\left|W_{y}\left(s_{i}^{*},b^{*}:b_{L}\right)\right|$.

By plotting $\ln\left|W_{y}\left(s_{i}^{*},b^{*}:b_{L}\right)\right|$ versus the time axis, a straight line is obtained for most of the time span of response because the modulus of CWT coefficients decays exponentially with time. This straight line is fit with a linear function, and the slope parameter is extracted. This slope parameter is equal to
(5)$$\frac{d}{dt}\ln\left|W_{y}\left(s_{i}^{*},b^{*}:b_{L}\right)\right|=-\xi_{i}\times \omega_{n}=\mathrm{slope},$$
where ξ is the damping ratio, and $\omega_{n}$ is the undamped natural frequency.

The relationship between the damped and undamped natural frequencies is well-known from structural dynamics as $\omega_{d}=\omega_{n}\times \sqrt{1-\xi^{2}}$. Hence, in practice, the damping ratio is obtained as
(6)$$\xi =\pm \sqrt{\frac{{\mathrm{slope}}^{2}}{\omega_{d}^{2}+{\mathrm{slope}}^{2}}}.$$

## 3. Damage Detection Algorithm

SHM systems estimate the health state of structures during operation to ensure their safety and economic efficiency. Such structures can be industrial, transport or energy equipment with structural elements, for example, wind turbines or wind generators and their elements—tower and blades. An SHM system’s sensors network will be connected to the structure, read the vibration data and send it to the workstation (computer). Then, an operator estimates the modal parameters of the structure. The resulting parameters together with the possible external operational factors, such as static loads, temperature or speed of rotation (blades), are input to a modal passport [21,22], where the past measurements are stored. Modal parameters in a reference or intact structural state together with their deviations due to operational factors form a “signature” of a structure that is unique to this structure or for structures of this type. As next step, a specialized algorithm analyses the modal passport and recognizes a particular structure. By analysing modal parameter changes with damage, one can infer on the severity of the damage, which allows for further planning of the agenda of structural serviceability—repair, replacement or resuming operation if damage is not significant.

The damage detection algorithm proposed for an SHM system originates from an anomaly detection field. It has three distinct phases, namely, Phase I: signal collection, Phase II: feature extraction, and Phase III: statistical control, as shown in Figure 1.

### 3.1. Phase I—Signal Collection

Vibration signals as a response to structural excitation are measured with N sensors connected to measurement channels Ch1 to ChN. In the case of numerous instances of impact excitation (p excitation of the structure during the measurement session), any individual free-vibration decay profile is isolated from the whole signal. Afterwards, p, these individual vibration profiles, are averaged to obtain N averaged vibration responses.

### 3.2. Phase II—Feature Extraction

In Phase II, a time-frequency analysis of the averaged responses is performed using CWT. CWT analysis involves the identification of wavelet ridges from the ridge condition in Equation (2) in the time domain signal analysed. Wavelet ridges represent the oscillatory modes, which comprise the components of vibration decay signals. Subsequently, this ridge information from the ridges identified at r=1,…, R is used to identify the resonant frequencies $f_{r}$ from wavelet phase (Equation (4)), and the decay profile of CWT coefficients in time domain is used to extract the damping ratios $\xi_{r}$ (Equation (6)). CWT analysis involves operating with wavelet scale parameters. In order to convert the scale parameter to frequency values, a wavelet scale-to-frequency conversion is realized through Equation (3) and is illustrated in Figure 2 for the analytical Morlet wavelet function. In this study, Morlet mother wavelet is used since it exhibits a high correlation with the time domain vibration data. This mother wavelet has been used for structural damage detection [23,24]. The identified modal parameter value pairs form features $f_{r}$ and $\xi_{r}$ that are organized into a two-column matrix $\left[\begin{array}{cc} f_{r} & \xi_{r} \end{array}\right]$, where rows correspond to observations and columns correspond to features. This feature matrix is used in the next phase of the anomaly detection algorithm proposed.

### 3.3. Phase III—Statistical Control

Phase III is concerned with performing statistical control of the feature values. The first stage is the feature-value filtering, which is carried out by using the interquartile range (IQR) rule. The goal is to remove the outliers from the features. Unlike the threshold set to mean plus/minus two or three standard deviations, the IQR approach for outlier removal is appropriate for data that does not necessarily follow a normal distribution. The frequency values for the ridges identified at r=1,…,R are filtered according to
(7)$$Q_{1}\left(f_{r}\right)-1.5\times \mathrm{IQR}\left(f_{r}\right)<f_{r}<Q_{3}\left(f_{r}\right)+1.5\times \mathrm{IQR}\left(f_{r}\right),$$
where IQR is the interquartile range, $Q_{1}$ is the first quartile, and $Q_{3}$ is the third quartile of the filtered resonant frequencies from the previous step. The filtered resonant frequency and damping ratio values are stored as two-column vectors $\left[\begin{array}{cc} f_{r}^{*} & \xi_{r}^{*} \end{array}\right]$.

The next step involves exploring the underlying probability distribution of the filtered features. For this purpose, the kernel density estimate (KDE) is computed. The reason is that a kernel distribution representation of the probability density function (PDF) of the data does not make any assumptions on the underlying distribution. The KDE is defined by a smoothing function and a bandwidth value that controls the smoothness of the resulting density curve. The kernel density estimator of the data at hand (x) is given by
(8)$${\hat{f}}_{h}\left(x\right)=\frac{1}{nh}\sum_{i=1}^{n}K\left(\frac{x-x_{i}}{h}\right),$$
where n is the sample size, K is the kernel-smoothing function governing the shape of the curve used to generate the PDF estimate, and h is the bandwidth. In this study, the obtained vectors of filtered frequency and damping ratio values are used as the data x, and normal density is used as a kernel smoothing function since the feature values approximately follow Gaussian distribution. It is important to choose the optimum bandwidth parameter since it regulates the degree of smoothing. In this work, bandwidth optimization was carried out by the following procedure illustrated in Figure 3:


Perform a cross-validation partition on the data to create 10 folds where one fold is used for testing and 9 folds are for training. Perform 10 iterations of such a partition, where a different fold is used for testing in each iteration.Define a range of bandwidth parameters b to test.In each training fold and the single testing fold, compute the KDE according to Equation (8) for each value of the bandwidth parameter. Then, compute an error ε between the KDEs of the testing and each training set according to
(9)$$\varepsilon_{b,\;k}\left(k-1\right)=\frac{1}{n}\sum_{i}^{n}\log{\hat{f}}_{h_{b}}\left(k-1\right),$$
where k=1:10 is the number of folds.Calculate the cross-validation error as a mean-squared-error of the errors in Equation (9) across all folds for each value b according to
(10)$${\mathrm{CVE}}_{b,\;\;k}=\frac{1}{k}\sum_{k}\varepsilon_{b,k}^{2}.$$Find the optimum bandwidth parameter by calculating the minimum of these cross-validation errors across all bandwidth values
(11)$$b_{opt,\;k}=b\left(\min\left({\mathrm{CVE}}_{b,k},\;b\right)\right).$$


Afterwards, the KDEs with these optimized bandwidth parameters are calculated for reference and all monitoring cases. Then, a centroid value of the KDE for both features is calculated according to
(12)$$C_{x}=\frac{\sum_{i}^{n}x_{i}\times {\hat{f}}_{h,\;i}}{\sum_{i}^{n}{\hat{f}}_{h,i}},$$
where data x in this study are the filtered feature values. Thus, both quantities, $C_{f_{r}^{*}}$ and $C_{\xi_{r}^{*}}$, are calculated for reference and all monitoring cases. Next, the centroid values of both features are organized into a row vector for each case at hand, and the Euclidean distances between centroids at reference and each monitoring case are calculated as
(13)$$d\left(C_{\mathrm{ref}},C_{\mathrm{case}}\right)=\sqrt{\sum_{i}^{m}{\left(C_{\mathrm{ref},i}-C_{\mathrm{case},i}\right)}^{2}},$$
where $C_{\mathrm{ref}}=\left[\begin{array}{cc} C_{f_{r}^{*},\mathrm{ref}} & C_{\xi_{r}^{*},\mathrm{ref}} \end{array}\;\right]$, $C_{\mathrm{case}}=\left[\begin{array}{cc} C_{f_{r}^{*},\mathrm{case}} & C_{\xi_{r}^{*},\mathrm{case}} \end{array}\;\right]$, and i=1, 2 since the centroid vector contains two values.

Once the Euclidean distance between the reference and all monitoring cases is calculated, a threshold value is established according to the following scheme:6.Consider all available structures of the same type at their reference state.7.Perform a modal parameters estimation to form feature vectors and calculate their centroid values for each structure.8.Calculate the Euclidean distance between centroid values in all possible combinations of structure pairs.9.Calculate the median value of these Euclidean distances and confidence bounds as

(14)$$\mathrm{CB}=Nq\pm z\sqrt{Nq\times \left(1-q\right)},$$
where N is the number of Euclidean distance samples, q=0.5 is the quantile corresponding to median (50% of data), and z is the critical value dependent on a chosen confidence level. For confidence level 0.95, z=1.96. The threshold of the Euclidean distances is set as a lower confidence bound at $T=Nq-z\sqrt{Nq\times \left(1-q\right)}$.

## 4. Experimental Campaign

The algorithm proposed is validated on the modal parameters extracted from five glass-fibre-reinforced polymer composite specimens with a cylindrical shape manufactured in the scope of an SHM system prototype research project. Cylindrical structures mimic the structural components of serial production, such as a helicopter tail boom, for which the current anomaly detection algorithm is intended.

### 4.1. Specimens

Testing objects are the structures in the form of cylinders fabricated from a composite material with the flanges made of plywood rings. The specimens are made of 300 g/m^2^ fibreglass fabric with a fibre orientation of 45° and LG 385 epoxy resin (HG 385 hardener). The weight of the specimen with the upper and lower flanges is 4.37 kg. Photo of a specimen is shown in Figure 4a. The specimen design includes (see Figure 4b):Item 1—composite cylinder made of fiberglass and epoxy resin;Items 2 and 3 —top and bottom annular flanges for cylinder fixation made of laminated plywood (30 mm thickness), respectively;Item 4—a network of 48 piezoelectric strain sensors;Item 5—wires connecting the sensors;Item 6—4 D-SUB type connectors at the places for connector fastening.

The dimensions of the specimen are as follows: nominal diameter of 300 mm, nominal length of 710 mm (with the flanges of 773 mm) and wall thickness of 1.45 ± 0.05 mm.

### 4.2. Measurement Subsystem

Each specimen includes its own sensor network, which outputs the signals during testing to the measuring system, providing a registration and storage of the signals.

The sensor network of each specimen comprises 48 polyvinylidene fluoride piezo film sensors connected to four conductor terminals and wiring harnesses with connectors. These films are flexible, lightweight and have piezoelectric properties [3]. Due to these properties, a strain in the film causes a change in stress. The piezo film is located between two printed silver electrodes, forming a capacitor-like structure. The dimensions of the sensors are roughly 45 mm × 20 mm × 0.05 mm, electrical capacity of 1.3 nF, operating temperature from −40 °C to 60 °C. The view of the sensor prepared for gluing on the specimen is shown in Figure 5a. The wires used for the SHM system prototype are the small, lacquered copper wires with a diameter of 0.25 mm. These wires are glued to the sensors with a special two-component epoxy glue, after which the sensor is covered with a nonconductive insulating tape. The electrical conductors connecting the sensors with the terminals (Figure 5b) are laid in the form of bundles in the longitudinal and circumferential directions and fixed with adhesive tape. On each terminal, 12 similar (conditionally signal) conductors are assembled on 12 contacts and 12 conditionally negative conductors on one common contact. Bundles of wires are soldered to the terminal contacts. The sensors are installed on the specimens in accordance with the premade markings, as shown in Figure 5c. At this stage, the sensor network is covered with a protective composite layer, and the specimen is glued into the annular grooves of the flanges.

### 4.3. Modal Testing

After the instrumentation installation, each specimen is fixed on a U-shaped modal testing stand, which, in turn, is mounted on a vibration isolation base (see Figure 6). Structural excitation is performed by repeated impacts of the specimen with a plastic modal hammer in the radial and vertical directions for 120 s. This test procedure is repeated 3 times. The Brüel & Kjær (B&K, Singapore) system LAN-XI Type 3053 is used as the data-reading device. A total of four Type 3053 modules were used to measure 48 channels simultaneously. A portable computer with software from the manufacturer of measurement modules (B&K, Singapore), such as Pulse Labshop, was used for data collection, processing and management. The entire signal recording with a duration of 20 s and sampling frequency of 4096 Hz contained 4 to 6 free vibration decay responses corresponding to the 4 to 6 instances of impact excitation. The test cases realized in the current study involve a reference state and progressing damage. The damage comprises a circular hole drilled through the thickness of the specimens in the same location. The diameter of the hole is increased and has the following values: 4, 8, 16, 24, and 32 mm.

All test specimens manufactured were visually inspected to check for defects or damage, as well as their compliance with the specifications. Local deviations and small differences between the specimens were identified to be mainly due to the factors of hand-made technology. These factors include uneven filling of the specimens and flange joint, resin leaks on the specimen surface, air bubbles in the protective layer, undercut on the flange of specimen No. 5, slightly different location of the first hole on the flange for each specimen, resin pouring out at different locations on the inner and outer surfaces of specimens, and differences in wire layouts for sensors.

### 4.4. Test Cases

The test cases realized in the current study involve reference state and progressing damage, namely, a circular hole drilled through the thickness of the cylinders in the same location. Diameter of the hole is increased in five stages as follows—4, 8, 16, 24, and 32 mm. The location of the hole schematically is shown in Figure 7a, while the close-up photo view of a 4 mm hole is shown in Figure 7b.

## 5. Results

### 5.1. Time-Frequency Analysis

An example of a response signal (specimen No. 1 from measurement channel 1) at a reference state is shown in Figure 8. The sampling frequency of the signal recording was 4096 Hz. This response signal was divided into separate free-vibration decay signals. The duration of each of these vibration signals was approximately 0.1 s. Afterwards, these responses were averaged and an averaged response for each measurement channel was obtained.

CWT scalograms at the reference state for the averaged time domain responses are shown in Figure 9. The scalogram is a useful analysis tool for signal analysis in joint time and frequency domains via the CWT. It is a 3D plot providing an indication of relative energy distribution across wavelet scales (related to frequencies) and time instants. Red regions mark the coordinates in time-frequency plane with high energy localization. Ridge identification using the ridge condition is shown on the bottom plots. The identified ridges and the corresponding frequencies are presented in Table 1. The frequencies correspond to the ones from the scale-to-frequency conversion from Figure 2 and do not consider variations associated with individual specimens. The vibration modes at scales 6 and 14, corresponding to 185.3 Hz and 106.4 Hz were identified for all five specimens. The vibration mode at scale 7 (172.9 Hz) was identified for all specimens, except for the second, while the vibration mode at scale 5 (198.6 Hz) was identified only for the first and fourth specimens.

### 5.2. Modal Parameter Estimation

#### 5.2.1. Resonant Frequencies

The estimation of resonant frequencies from wavelet phase is shown in Figure 10a. First, CWT coefficients were calculated from the averaged vibration response. Second, phase angle between the real and imaginary CWT coefficients were calculated separately for each identified wavelet ridge. Resonant frequencies were calculated as a derivative of wavelet phase according to Equation (4). The number of identified observations of resonant frequency increased with increasing the signal length and sampling frequency. Therefore, it was possible to extract large sample size of features from a long signal with high sampling frequency.

#### 5.2.2. Damping Ratio

The procedure of damping ratio estimation is shown in Figure 10b. First, the moduli of the CWT coefficients was calculated at each wavelet ridge along the time axis. Second, the natural logarithm was taken, and this result was approximated with a linear function. The slope of this fit was recorded, and the damping ratio was calculated according to Equation (6).

#### 5.2.3. Frequency Filtering

The results of the frequency filtering for the vibration mode at 185 Hz (scale 6) at a reference state for specimen No. 2 are illustrated using histograms in Figure 11 and shown numerically in Table 2. There are significant outliers in the extracted instantaneous frequencies for all specimens. The small inset shows a histogram of the zoomed-in portion before filtering the histograms, ignoring the outlier values. The histograms on the right show the results after filtering. The bimodal character of frequency distribution can be traced. The results in Table 2 show that while the mean frequency values change only slightly, the standard deviation and, especially, frequency range reduced drastically after the filtering process.

### 5.3. Kernel Smoothing

After feature filtering, the next step is computation of the KDE of the filtered features. It was shown that feature values do not follow a clear normal distribution; therefore, KDE is an appropriate tool for the estimation of the underlying probability density. In the bandwidth optimization routine, bandwidth values were set from 0.01 to 1, with a total of 100 values for testing. The bandwidth parameter optimization results for specimens at the reference state are shown in Figure 12a. It can be seen that the optimum bandwidth parameters are different for different specimens even if they are designed to be equal. The range of optimum bandwidths is from 0.03 (specimen No. 2) to 0.14 (specimen No. 4). The optimized KDE of the resonant frequency distribution at scale 6 for the specimen No. 1 at reference in comparison to a KDE obtained with a default bandwidth parameter is shown in Figure 12b. The optimization procedure removes the spikes of the KDE producing a smooth curve.

The optimized KDEs for a specimen No. 1 are shown in Figure 13. Here, all damage scenarios are shown along with the reference for both resonant frequency and damping ratio features. Centroids are marked as filled circles of a colour corresponding to one of the associated KDE. The KDEs reveal complex multimodal distributions, indicating that the underlying probability densities are, in fact, not classical Gaussian for all damage cases.

### 5.4. Damage Detection

#### 5.4.1. Threshold Estimation

The threshold values for the Euclidean distances of centroids at reference are estimated from all five cylindrical specimens. Hence, the total number of combinations is 10 if that particular vibration mode is identified for all structures. The Euclidean distances calculated for three modes of vibration at 185, 176 and 106 Hz are shown in Figure 14. Only six combinations for vibration mode at 176 Hz were obtained meaning that this particular mode was identified in four out of five specimens. The values of the Euclidean distances are not uniform, indicating a relatively high standard deviation, particularly for modes at 185 and 106 Hz. Overall, the largest deviations from the reference are for the vibration mode at 185 Hz, while the smallest are the for vibration mode at 176 Hz.

The median values and their confidence bounds according to Equation (14) of the calculated Euclidean distances are presented in Table 3 for all vibration modes. The threshold value is set as a lower confidence bound to ascertain that the structural change is detected.

#### 5.4.2. Damage Indication

The damage detection results are displayed in Figure 15. Red horizontal line shows the threshold level. The scales for the Euclidean distances for each mode of vibration are set to the maximum value among all five specimens to compare the magnitude of deviation at the reference among different specimens. First of all, it can be seen that the values of the Euclidean distances are increasing with progression of damage, indicating that the approach proposed is sensitive to changes in damage severity. Secondly, the largest overall deviations from the reference state are observed for features from the vibration mode at 106 Hz. Furthermore, the values of Euclidean distance for all damage scenarios are above the threshold level for all specimens. From this perspective, the vibration mode at 185 Hz yields the poorest results since only the most severe damage cases are detected (above the threshold) for specimens Nos. 1, 2 and 3. No damage for this vibration mode is detected for specimens Nos. 4 and 5. Almost all cases of damage were detected for the vibration mode at 176 Hz. These observations indicate that it is crucial to consider multiple modes when detecting damage since, firstly, there might be missed detections if features were extracted only from a single mode and, secondly, the features of different vibration modes display different magnitudes of deviations from the reference.

Limitations of the algorithm proposed are directly associated with the quality of the vibration signals recorded, which, in turn, strongly depend on the noise robustness of the sensors to the environmental conditions in which the structure is operating. Secondly, the algorithm does not consider the operational deflection shapes (ODSs) as an additional feature. It is anticipated that information on the ODSs could potentially enhance the discriminative power of the algorithm. It is possible to identify the ODSs with continuous wavelet transform. However, it would significantly increase the complexity of the algorithm, which was not the aim of this study. Thirdly, damage detection accuracy depends on an accurate threshold estimation. Obviously, according to statistics, a larger sample of structures would have yielded a more accurate threshold value for the Euclidean-distance damage indicator. However, there is a trade-off of accuracy and number of structures to be manufactured for such threshold calculations involving increased financial and time resources.

#### 5.4.3. Comparison with Mahalanobis Distance

The damage identification results obtained were compared to the well-known Mahalanobis distance (MD) metric. The MD values follow a chi-squared (*χ*^2^) distribution [25,26]. Thus, threshold for the MD values is defined as an inverse of *χ*^2^ cumulative distribution function with *υ* degrees of freedom and a selected probability P: $T_{MD}=\sqrt{\chi_{\nu ,P}^{2}}$. The results of damage detection for specimen No. 1 are shown in Figure 16. The threshold level is selected at 95% probability and two degrees of freedom (corresponding to the two feature vectors in a feature matrix). It can be seen that while the majority of the green dots (reference MD data) lie under the threshold, the damage MD data are largely above the threshold for the vibration modes at 106 and 176 Hz. On the other hand, a poor damage detection can be seen for the vibration mode at 185 Hz, since a relatively small proportion of the damage MD data exceed the threshold.

The evaluation power of both damage indicators is estimated through the false alarm rate (FAR).
(15)$$\mathrm{FAR}=1-\mathrm{Accuracy}=1-\frac{\sum_{i=1}^{m}{\left(\mathrm{MD}>T_{\mathrm{MD}}\right)}_{i}}{\sum_{i=1}^{m}{\mathrm{MD}}_{i}},$$
where m is the number of observations of the Mahalanobis distance metric for each case. The accuracy, on the other hand, is 1−FAR and is presented in Figure 17. The colourmaps show the accuracy (in %) of damage detection for the Euclidean distance metric (on the left) and Mahalanobis distance metric (on the right). The colour corresponds to the accuracy percentage level. For the Euclidean distance, there can be either 0% or 100% accuracy (Euclidean distance either does not reach the threshold or cross it), while for the Mahalanobis distance, any intermediate accuracy can be achieved since numerous observations of this metric are obtained. Additionally, accuracy for a reference state for the Mahalanobis distance can be assessed, which is not the case for Euclidean distance. NaN (not a number) means that the vibration mode was not identified.

The vibration mode at 185 Hz is the least favourable for damage detection (accuracies are low for both methods), while both other modes are roughly equal in this regard. Overall, the accuracies are comparable between both methods. In cases where there is 100% accuracy for the Euclidean distance, the corresponding accuracy for the Mahalanobis distance is usually lower, since not all observations have passed the threshold. The advantages of the method proposed are that computation for the Euclidean distance is simpler and requires less computational power since the covariance matrix does not need to be computed.

## 6. Conclusions

In the current study, an anomaly detection algorithm based on output-only structural vibration responses of structural components was proposed. The algorithm detects changes of modal parameters caused by structural degradation, such as the progression of damage. Phase I deals with the acquisition of vibration response signals from the sensors mounted on the structure and subsequent signal averaging and fusion. Phase II uses the modal parameters as the features identified with continuous wavelet transform routine. Phase III is carried out for decision-making regarding the state of integrity of the structure in question based on the statistical descriptors of the extracted features. Here, the feature filtering scheme based on IQR Rule is adopted to remove outlier feature values. Feature filtering revealed that the distribution for some specimens is, in fact, bimodal, while for others it is skewed and not classical Gaussian. The probability density function of the filtered features is estimated using a kernel density estimate (KDE) to avoid the assumption that the underlying distribution is normal. The damage indicator is based on the Euclidean distance between the centroid of KDE for both features at reference state and any state of damage. The following can be concluded:The Euclidean distance of the centroids of the modal features KDEs between the reference and damage states can be used to detect damage.The damage indicator proposed shows an upward trend for damage progression, meaning that it is effective in detecting increasing severities of damage.Some vibration modes are more sensitive to damage than others. Therefore, multiple vibration modes have to be identified in order to increase the reliability of the damage detection. For example, features originating from the vibration modes at 106 and 176 Hz have significantly higher deviations from the reference than the vibration mode at 185 Hz. Therefore, these vibration modes were more effective in damage detection. On the other hand, these vibration modes could not be identified for all damage cases, while the vibration mode at 185 Hz was present in all scenarios.There is a significant scatter of the feature value deviations from reference among the test samples. For the most part, this is due to inconsistencies in the sample design and instrumentation, as mentioned in Section 4.1. Specimens.The damage indicator proposed was compared to the Mahalanobis distance metric for damage detection. Both methods yield comparable damage detection accuracy. Therefore, there is no reason to use a more computationally costly Mahalanobis distance approach.

Future research will be devoted to the consideration of an influence of environmental and operational factors on the structural characteristics and modal parameters for a practical SHM system. To reduce the influence of the ambient temperature, for example, some papers consider methods for suppressing this effect as interference [27], in others, methods for constructing quantitative models, that accurately predict the modal frequency that corresponds to temperature change are proposed [28]. A promising way to consider influence factors is to utilize the modal passport mentioned above, which is a method for collecting and processing all modal data of a structure while taking into account environmental and operational variances.

## Figures and Tables

**Figure 1 sensors-23-06121-f001:**
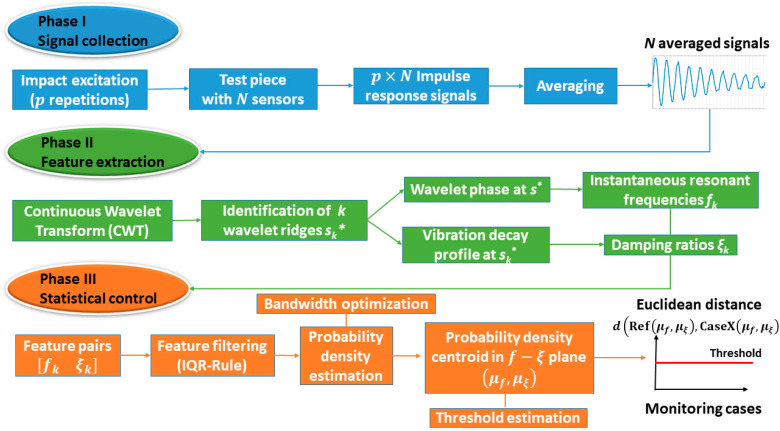
Anomaly detection algorithm.

**Figure 2 sensors-23-06121-f002:**
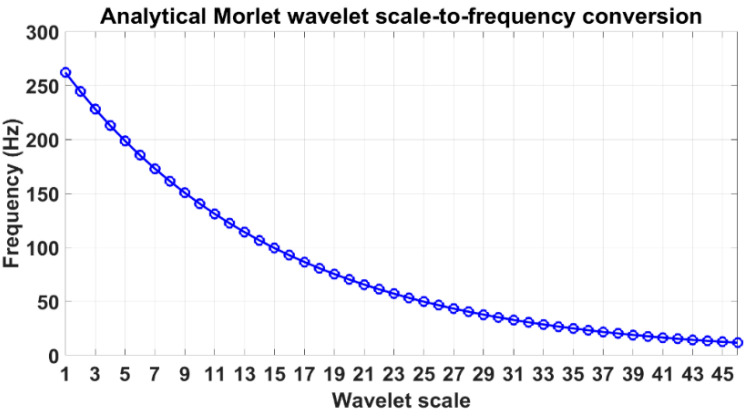
Wavelet scale-to-frequency conversion plot for the analytical Morlet wavelet.

**Figure 3 sensors-23-06121-f003:**
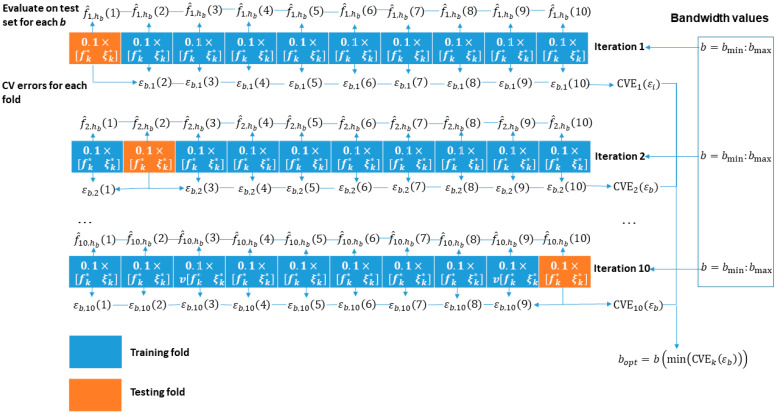
Optimization scheme for the kernel density bandwidth parameter.

**Figure 4 sensors-23-06121-f004:**
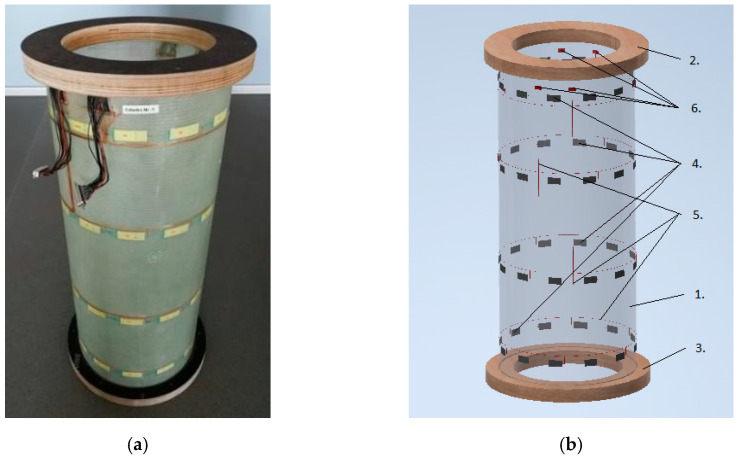
Specimen test object No. 1: (**a**) general view; (**b**) design.

**Figure 5 sensors-23-06121-f005:**
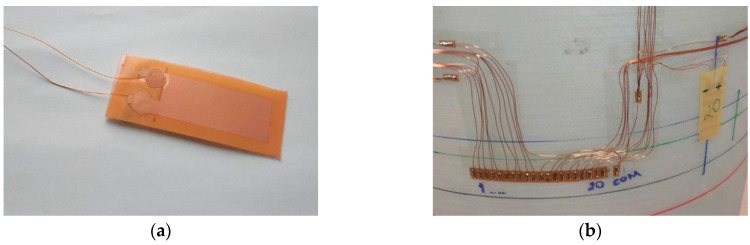
Installation of sensor system on the test specimens: (**a**) piezoelectric film sensor used in the measurements; (**b**) sensor terminal glued around the circumference of the cylinder; (**c**) arrangement of a sensor network on the specimen.

**Figure 6 sensors-23-06121-f006:**
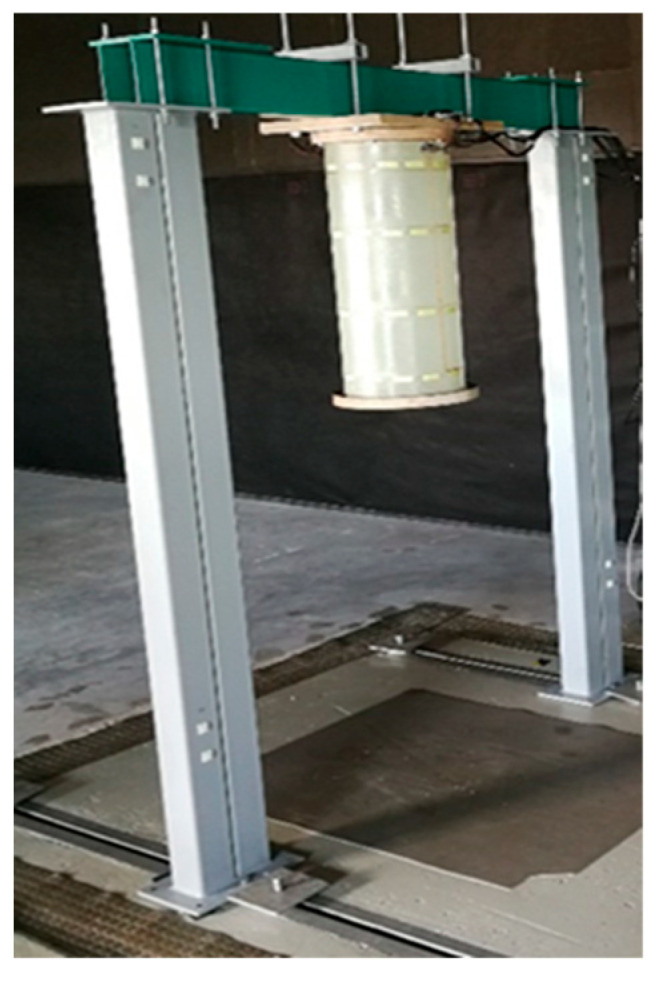
Modal test stand.

**Figure 7 sensors-23-06121-f007:**
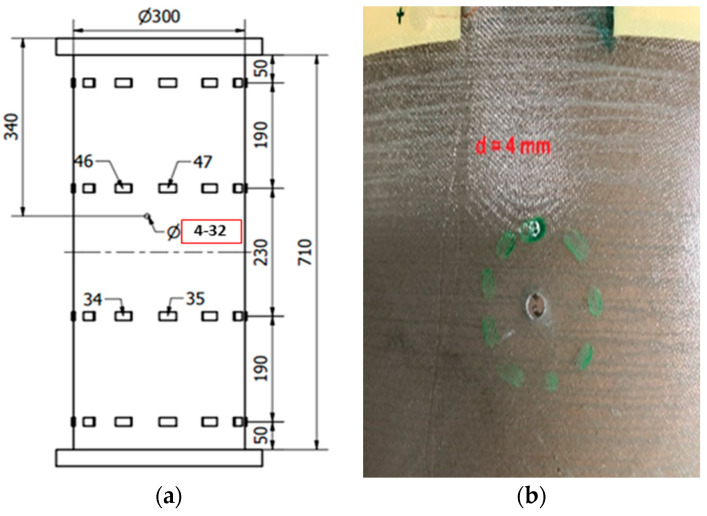
Damage in the test specimens: (**a**) schematic showing the location and size of the hole; (**b**) photo of the 4 mm hole. The hole is located close to sensors No. 46 and 47.

**Figure 8 sensors-23-06121-f008:**
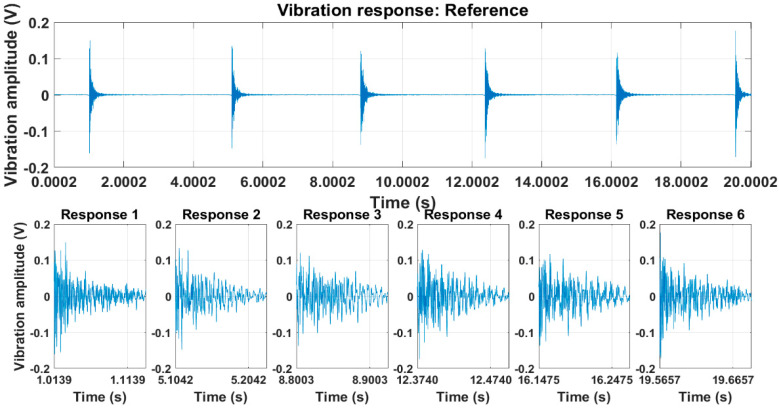
Recorded free vibration decay (specimen No. 1, measurement channel 1).

**Figure 9 sensors-23-06121-f009:**
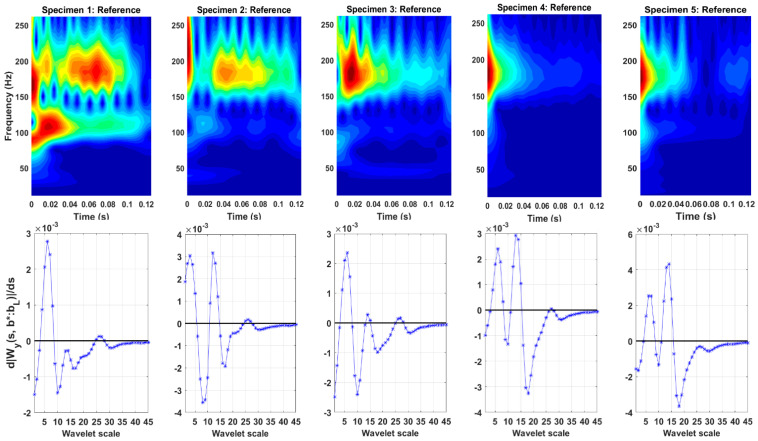
Resonant frequency identification with CWT. Top: CWT scalograms showing signal energy distribution at different frequencies and time instants; bottom: derivative of moduli of CWT coefficients versus wavelet scale. Scales at zero crossings (thick black line) corresponding to wavelet ridges. Note the inverse proportionality of frequencies and wavelet scales.

**Figure 10 sensors-23-06121-f010:**
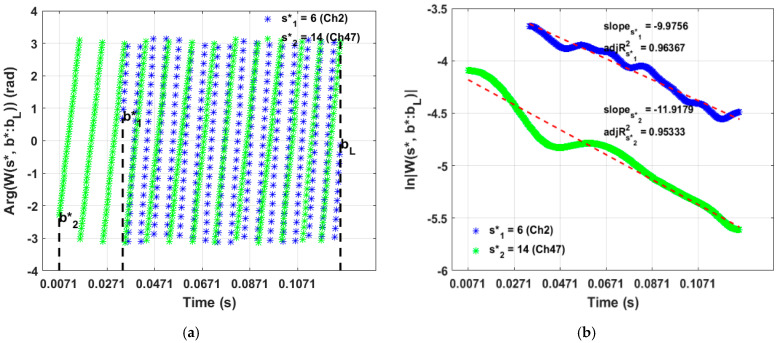
Estimation of modal parameters from wavelet ridges: (**a**) CWT of the signal and estimation of resonant frequencies from wavelet phase (two ridges shown); (**b**) extraction of slope of decay related to damping ratio.

**Figure 11 sensors-23-06121-f011:**
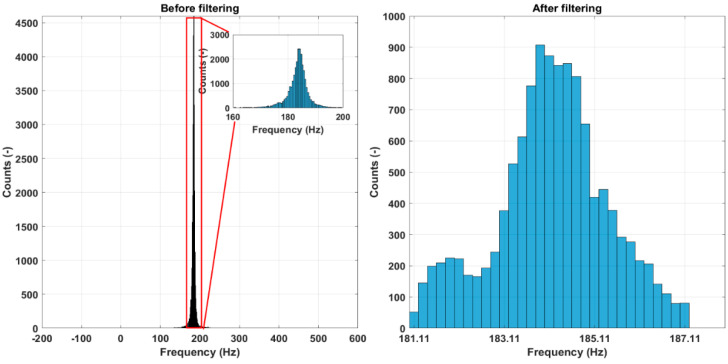
Resonant frequency filtering for a reference state, vibration mode at 185 Hz, specimen No. 2.

**Figure 12 sensors-23-06121-f012:**
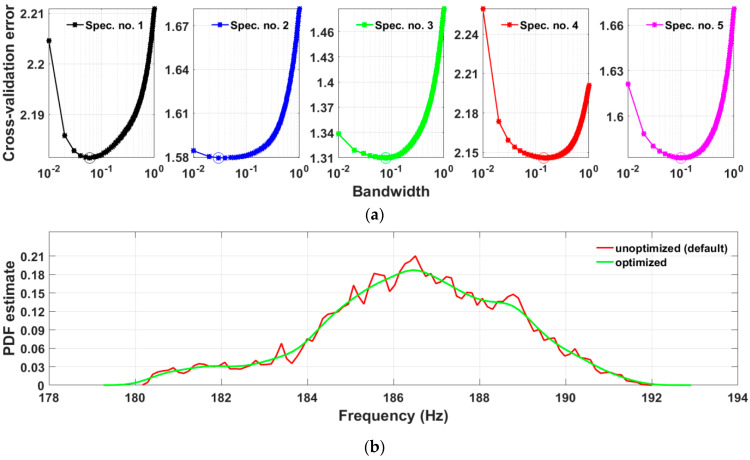
Bandwidth optimization. (**a**): cross-validation error for a range of bandwidth parameters where minimum value is marked with a circle; (**b**): kernel density estimate of probability density function for the optimized bandwidth parameter and default (specimen No. 1 at reference state). Optimization of bandwidth has removed the spikes of KDE smoothing out the curve.

**Figure 13 sensors-23-06121-f013:**
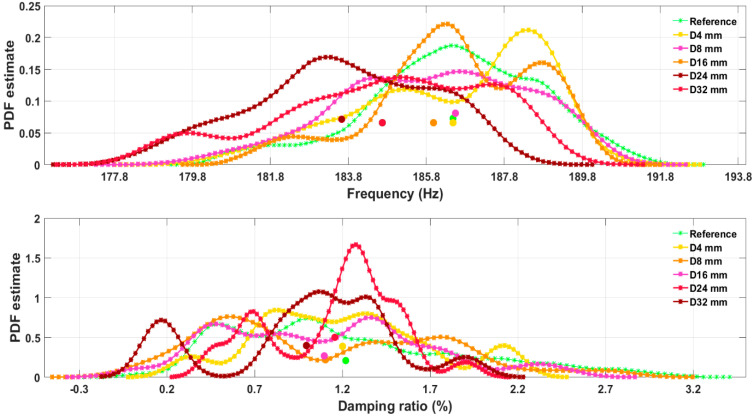
Kernel density estimate for specimen No. 1 at reference and damage states for resonant frequency and damping ratio modal features. Centroids are marked with filled circles. Their position with respect to the reference can be seen.

**Figure 14 sensors-23-06121-f014:**
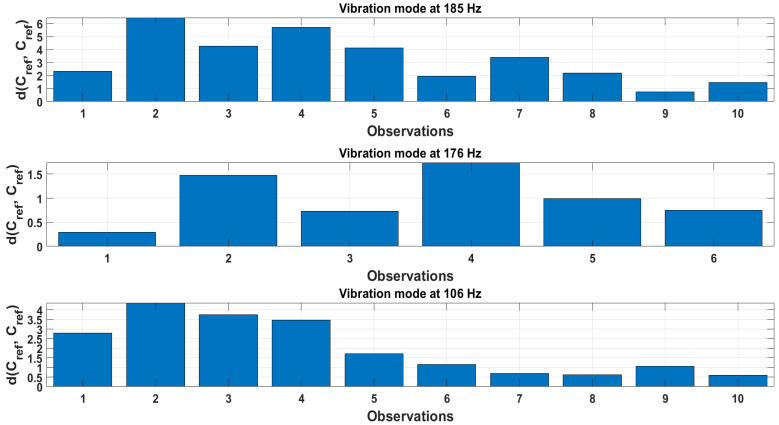
Euclidean distance values of centroids between reference states of all structures for the extracted vibration modes.

**Figure 15 sensors-23-06121-f015:**
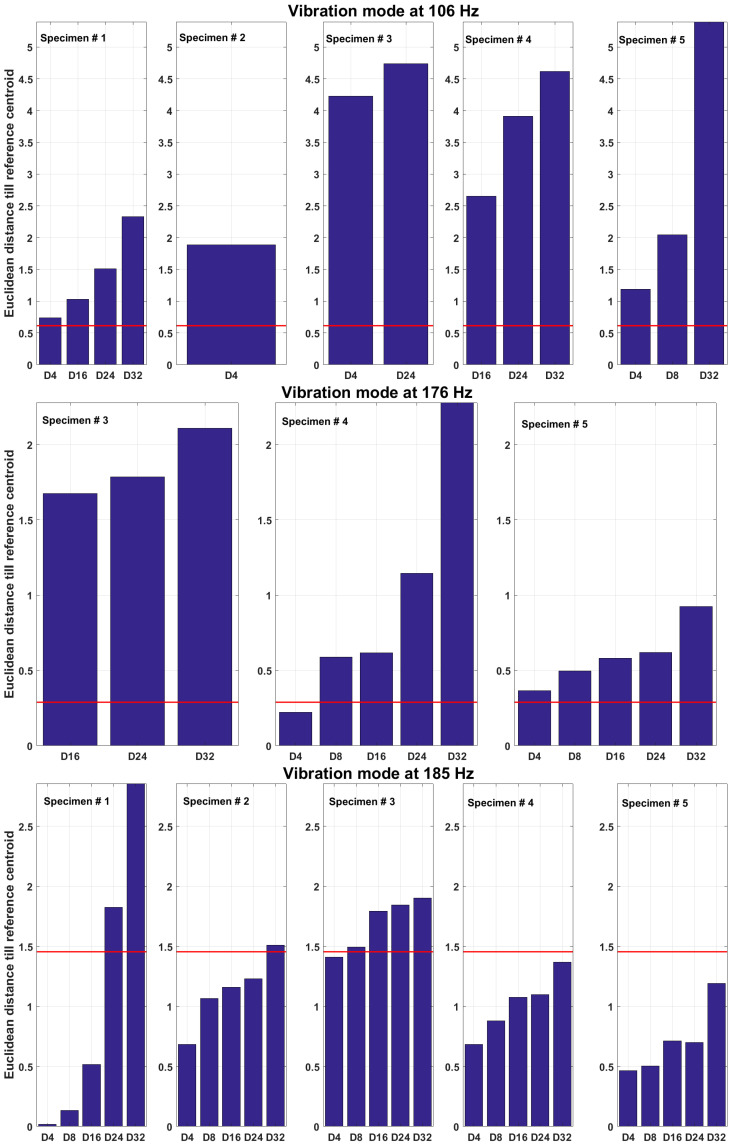
Damage index as Euclidean distance between feature centroids at reference state and damage states. Red horizontal line shows the threshold for each mode of vibration.

**Figure 16 sensors-23-06121-f016:**
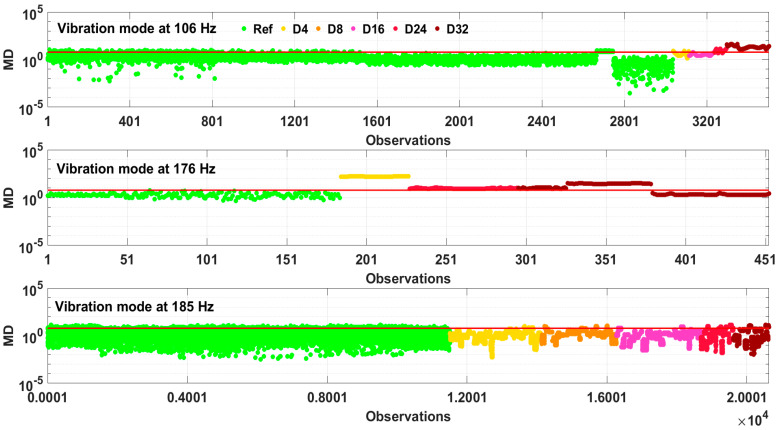
Mahalanobis distance (MD) metric in logarithmic scale for the identified vibration modes for specimen No. 1. Red line shows the threshold level. Different vibration modes have a different damage detection performance.

**Figure 17 sensors-23-06121-f017:**
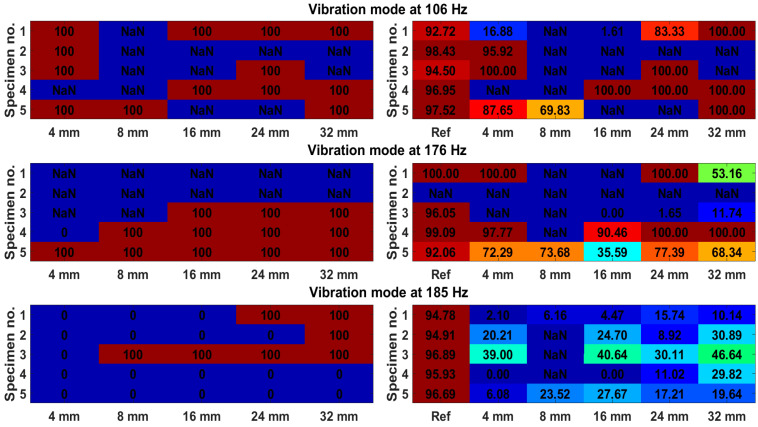
Colormaps of accuracy of Euclidean distance metric (**left**) and Mahalanobis distance metric (**right**) for three vibration modes. NaN means that the vibration mode was not identified for that particular case of damage.

**Table 1 sensors-23-06121-t001:** Wavelet scales and frequencies of the identified wavelet ridges of the averaged response signals.

	Specimen	1	2	3	4	5
Ridge #	Scale *s* (-)	*f* (Hz)	*f* (Hz)	*f* (Hz)	*f* (Hz)	*f* (Hz)
1	5	198.6	-	-	198.6	-
2	6	185.3	185.3	185.3	185.3	185.3
3	7	172.9	-	172.9	172.9	172.9
4	14	106.4	106.4	106.4	106.4	106.4

**Table 2 sensors-23-06121-t002:** Statistical descriptors of resonant frequency feature filtering.

Specimen No.	1	1	2	2	3	3	4	4	5	5
Filtering	No	Yes	No	Yes	No	Yes	No	Yes	No	Yes
Mean (Hz)	187.38	186.49	183.58	184.15	178.41	180.03	182.38	182.21	142.70	180.77
Variance (Hz^2^)	359.10	4.97	99.40	1.54	421.48	0.94	364.81	5.29	1474.56	1.49
Range (Hz)	1512.70	11.48	701.68	6.08	2501.22	4.37	2061.54	10.86	872.45	6.02

**Table 3 sensors-23-06121-t003:** Threshold estimation results for each vibration mode.

Vibration Mode	Scale 14 (106 Hz)	Scale 7 (176 Hz)	Scale 6 (185 Hz)
$\mathrm{median}\left(d\left(C_{ref},C_{ref}\right)\right)$	1.432	0.871	2.863
CB_lower	0.614	0.289	1.455
CB_upper	3.464	2.34	4.274

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
