# Peer review of "Wavelet-Based Output-Only Damage Detection of Composite Structures"

_sensors, 2023, doi:10.3390/s23136121_

Round 1
Reviewer 1 Report
Dear authors
After reviewing the paper, it can be concluded that the paper was well writen, organized, and presented. Howeverm there are few issues need to be adressed.
1) In abstract, the "parameters" in line 8, 10 should be declared in detail. Meanwhile, in line 17, the "proposed damage indicator" also need to be described clearly.
2) In introduction section, The literature review should be widen by adding more literatures to make readers know not only the theory but also the application;
3) In page 5, from line 189 to 199, the No. type should be not the same with first class title.
4) the figure 1, 2, and 3 should be right after the citation.
5) In Fig. 4, each part of the sample and installation should be indicated in the photos. The sensor type, device, and software applied, and settings should also be indicated.
6) Figrue 11, The title of x axis overlays the number. Please also check other figrues.
7) The line 423 to 434, should be moved to results and dicussion section.
Author Response
- Elaborated on modal parameters as resonant frequencies, damping ratios and operational deflection shapes. Proposed damage indicator is based on Euclidean distance. These changes are in blue color of the revised version.
- We have added 5 more reference sin the Introduction section regarding the use of wavelet transform for modal parameter estimation of real structures. These changes are in blue color of the revised version.
- Sorry, the authors did not understand this comment.
- Figures 1,2 and 3 have been moved right after the references to them in the revised version.
- A description of specimen instrumentation with sensors, wiring, etc. has been added along with 3 supplementary figures in the revised version. These changes are in blue color of the revised version.
- Figures have been remade and this fault has been corrected in the revised version.
- This portion of the text has been moved to Results section in the revised version.

Reviewer 2 Report
A review of
Wavelet-based Output-Only Damage Detection of Composite Shell Structures
by Rims Janeliukstis and Deniss Mironovs
This manuscript reports a wavelet-based method for damage detection of composite shell structures. The new algorithm utilizes kernel density estimate (KDE) of an underlying probability distribution and Euclidean distance between centroids of KDEs at reference and various other cases of structural integrity as an indicator of damage. In general, this paper is well-written. The following are my suggestions which may help the authors in further improving the quality of the paper:
1. Since the title of this study suggest that this method is for damage identification of composite shell structures, it is supposed that state-of-art of the damage identification techs regarding the composite structure shall be reviewed in the part of literature review. However, the current version only reviews the modal analysis methods for a broad range of structures. It seems that this method is not specially designed or modified for composite structures.
2. How damage is simulated in the experimental study? This is not clearly explained in the section of experimental campaign.
3. Details of the sensors used in the experimental study shall also be provided? What type of sensors are used? Strain? Displacement sensor? Where are the sensors installed? Any filtering technique used in this study?
4. It seems to me that Figure 5 illustrate the response of a specimen subjected to impact load. If so, this method cannot be named “output-only damage detection”.
5. Also, in Figure 5, the unit of the Y axis is volt, which contains no information to me. The obtained response shall be strain, stress, acceleration or displacement, etc.
6. What is the feature of the proposed wavelet-based method? A comparison with other method in accuracy or other performance is suggested to be added to this study.
Minor comments
1. “Figure 6” in Page 11 shall be “Figure 7”.
2. “Phase II” in Page 4 Line 166 shall be “Phase III”
Author Response
- The word "shell" is removed from the title as, indeed, this study doesn't focus on shell structures in particular. All changes are in blue color.
- We have added a subsection Test cases with 2 figures describing and showing the location of hole in a photo.All changes are in blue color.
- A description of specimen instrumentation with sensors, wiring, etc. has been added along with 3 supplementary figures in the revised version. These changes are in blue color of the revised version. All changes are in blue color.
-
It is true that specimen is being subjected to impact load, but that does not contradict the definition of output-only damage detection. The reason this method is called output-only, is that we do not know the input force, and we only consider output vibrational responses for analysis. This is why it is called output-only damage detection.
-
The vibrational response shown in Fig. 5 is an electrical signal of deformation from one of the piezo film sensors. The reason these values are not transformed into strain values is – the deformation sensors are not calibrated sensors and their sensitivity is unknown. This still allows for damage detection, as we operate with the obtained modal parameters – frequency and damping, and not with amplitude of the recorded signal. The modal parameters do not change from changing the units of the signal.
- A new subsection of Comparison with Mahalanobis distance has been added in the revised version. Here, Mahalanobis distance metric is also calculated, accuracies of the proposed method and Mahalanobis distance metric are compared and differences highlighted. All changes are in blue color.
- Namings of Figures and Subsections has been revised. All changes are in blue color.

Reviewer 3 Report
General considerations:
· The topic matter of the proposal addresses a very interesting problem; it deserves to be considered for a high impact publication.
· The document includes a total of 22 references, of which 57% are publications produced in the last 5 years, 26% in the last 5-10 years, and 17% are more than 10 years old, implying a total percentage of 83 % recent references. In this way, the total number of references used can be considered appropriate.
· An English review should be performed as there are many grammatical problems when dividing words between lines as they break syllables in the words.
· The research design is appropriate, the methodology is properly described, and the results and conclusions are clearly stated.
An English review should be performed, as there are many gramatical problems when dividing words between lines, as they break the syllables
Author Response
1. This is a feature of MS Word and probably the official template of the journal. The breaking of words to fit the lines is done automatically and authors cannot really affect this.

Reviewer 4 Report
The authors proposed an algorithm based on the continuous wavelet transform to identify damage parameters for the purposes of the structural health monitoring. The paper is recommended for publication after a certain correction.
1. It is not clear how the algorithm works if some resonance frequencies are not detected in some signals. Is it sufficient for the method or leads to some corrections? Please speicify.
2. Why the Morlet mother wavelet has been tested? Did the authors considered other kinds of wavelet such as Gabor wavelet and others?
3. Which time step and length of the signals are used in the study for the numerics during the application of the continuous wavelet transform and which is required for reasonable accuracy of resonance frequency estimation?
Minor editing of English language required only.
Author Response
- The method works by identifying wavelet ridges, which correspond to oscillatory signal components or modes. From this ridge information, resonant frequencies and damping ratios are extracted for each mode. As long as at least one wavelet ridge is identified, features for damage detection will be available. However, it is possible that the identified mode is not sensitive to damage. For example, it may have a zero amplitude at the location of damage. In that case, it is required to identify several modes so that the information on damage is complimentary.
-
Morlet wavelet has been used, since it is a popular approach to use Morlet wavelet for structural damage detection/localization problems involving vibration signals. Moreover, in our study Morlet mother wavelet had the highest correlation with the input response signals yielding the wavelet transform coefficients with the highest magnitude.
-
In our study, the signals were recorded with a sampling frequency of 4096 Hz. As can be seen in Figure 5 (Figure 8 in the revised version), the duration of a single measurement session has been 20.002 seconds containing 6 impact responses. These responses had been averaged and a single response with 512 values with a time step of 2.4414e-04 seconds for each sensor channel was obtained. This time step corresponds to frequency resolution of 1.5784 Hz for the lowest frequencies (around 22 Hz) and frequency resolution of 101.0182 Hz for the highest frequencies (around 1500 Hz). In our case, the frequencies of interest are from 106 Hz to 185 Hz, meaning that frequency resolution in this interval varies from 7.2525 Hz to 12.6273 Hz.

Round 2
Reviewer 4 Report
Accept.